# Near Infrared Spectroscopy for Muscle Specific Analysis of Intensity and Fatigue during Cross-Country Skiing Competition—A Case Report

**DOI:** 10.3390/s21072535

**Published:** 2021-04-04

**Authors:** Thomas Stöggl, Dennis-Peter Born

**Affiliations:** 1Department of Sport and Exercise Science, University of Salzburg, 5020 Salzburg, Austria; 2Red Bull Athlete Performance Center, 5020 Salzburg, Austria; 3Department for Elite Sport, Swiss Federal Institute of Sport, 2532 Magglingen, Switzerland; Dennis.Born@swiss-aquatics.ch; 4Swiss Swimming Federation, 3401 Bern, Switzerland

**Keywords:** double poling, GPS, GNSS, heart rate, muscle oxygenation, NIRS, Vasaloppet

## Abstract

The aims of the study were to assess the robustness and non-reactiveness of wearable near-infrared spectroscopy (NIRS) technology to monitor exercise intensity during a real race scenario, and to compare oxygenation between muscle groups important for cross-country skiing (XCS). In a single-case study, one former elite XCS (age: 39 years, peak oxygen uptake: 65.6 mL/kg/min) was equipped with four NIRS devices, a high-precision global navigation satellite system (GNSS), and a heart rate (HR) monitor during the Vasaloppet long-distance XCS race. All data were normalized to peak values measured during incremental laboratory roller skiing tests two weeks before the race. HR reflected changes in terrain and intensity, but showed a constant decrease of 0.098 beats per minute from start to finish. Triceps brachii (TRI) muscle oxygen saturation (SmO_2_) showed an interchangeable pattern with HR and seems to be less affected by drift across the competition (0.027% drop per minute). Additionally, TRI and vastus lateralis (VL) SmO_2_ revealed specific loading and unloading pattern of XCS in uphill and downhill sections, while rectus abdominus (RA) SmO_2_ (0.111% drop per minute) reflected fatigue patterns occurring during the race. In conclusion, the present preliminary study shows that NIRS provides a robust and non-reactive method to monitor exercise intensity and fatigue mechanisms when applied in an outdoor real race scenario. As local exercise intensity differed between muscle groups and central exercise intensity (i.e., HR) during whole-body endurance exercise such as XCS, NIRS data measured at various major muscle groups may be used for a more detailed analysis of kinetics of muscle activation and compare involvement of upper body and leg muscles. As TRI SmO_2_ seemed to be unaffected by central fatigue mechanisms, it may provide an alternative method to HR and GNSS data to monitor exercise intensity.

## 1. Introduction

In long-distance cross-country skiing (XCS), the nature of mass start races provokes rapid changes in exercise intensity due to drafting [1] and sprints for optimal positioning within the pack [2]. Additionally, these races usually take place in undulating terrain and exercise intensity increases during uphill sections, while downhill sections provide short periods for recovery [3].

Heart rate (HR) is commonly used to determine exercise intensity in endurance events due to low costs and high feasibility [4]. However, during XCS, varying exercise intensity with a high anaerobic energy contribution during steep uphill sections may overestimate oxygen consumption and energy expenditure estimated from HR [5]. Additionally, latency in the HR response to changing exercise intensity further limits its application in undulating terrain [6,7]. Moreover, cold temperature [8] and fatigue during long lasting endurance events reduce [9], while dehydration increases HR [10]. Those conflicting effects may all take place during a single XCS race and HR may not accurately reflect the metabolic demand and exercise intensity.

Velocity and changes in altitude measured with global navigation satellite systems (GNSS) may rule out some of the disadvantages, in particular latency of HR response and thermoregulatory effects [11]. However, velocity in XCS is affected by varying terrain, equipment, and environmental factors, i.e., glide wax, grip wax, base grinding structure, and ski stiffness, as well as air temperature, snow condition, humidity, radiation, and dirt in the snow [2]. Therefore, long-distance XCS demands load monitoring beyond HR and GNSS measurements.

Evolution of modern sensor technology towards miniaturized and wireless near infrared spectroscopy (NIRS) devices may provide a feasible solution for applications in the field and during real race scenarios. Muscle oxygenation as measured with NIRS has been shown to reflect oxygen uptake more accurately, and respond faster to changing exercise intensity when compared to HR [7,12]. As muscle oxygenation and deoxygenation can be used to determine metabolic threshold intensities during routine laboratory procedures, i.e., incremental step tests [13,14], NIRS can be used to monitor individualized intensity zones during training and competition. Additionally, costs of NIRS dropped from 4000–25,000 dollars in 2000 [15] to <1000 in 2020 (i.e., Moxy Monitor), which makes the technology accessible to a broad range of athletes and applicable beyond scientific research. 

As the various XCS techniques generally involve a large number of muscles of the whole body serving different functions (e.g., for propulsion versus stabilization) [16], and long-distance XCS races involve large portions of double poling (DP) with high demand on upper body and trunk muscles [9], muscle specific intensity may differ from systemic central intensity and fatigue mechanisms. Therefore, NIRS may provide more detailed knowledge when comparing local metabolic demand and intensity between different muscle groups [17,18]. However, robustness, non-reactiveness, and practicality in the field during competitions in rough conditions, i.e., cold temperatures and high-intensity whole-body efforts, are yet to be determined. 

Therefore, the aims of the present preliminary and explorative study were to (1) investigate contribution of various upper body, lower body, and trunk muscles to DP during a 90-km long-distance XCS race; (2) compare exercise intensity between HR, GNSS data, and local muscle oxygenation (NIRS signal) of the triceps brachii (TRI), latissimus dorsi (LD), rectus abdominus (RA), and vastus lateralis (VL) muscles with data expressed relative to laboratory all-out efforts; (3) investigate robustness and non-reactiveness of wearable NIRS technology during a real race scenario, such as a mass start long-distance XCS race with rapidly changing exercise intensity based on the varying terrain.

## 2. Materials and Methods

### 2.1. Participant

One former elite male Caucasian long-distance skier was recruited (age: 39 years, body mass: 80.4 kg, body height: 187 cm, maximal oxygen uptake (VO_2max_) DIA: 65.6 mL/kg/min, DP: 62.8 mL/kg/min) with approximately 25 years of experience in ski racing at the national and international level (World Cup, World Championships). The participant was fully informed about all study details and participation requirements, giving informed consent to participate. The study received approval from the local Ethical Committee (EK-GZ: 05/2017) and was conducted in accordance with the Declaration of Helsinki.

### 2.2. The Vasaloppet Race

The ‘Vasaloppet’ is one of the oldest and longest XCS races and with little doubt is the most famous in the world. The course is 90 km with a total altitude gain of 1380 m. It consists of various flat sections, sections with slightly undulating terrain, and two noteworthy hills (Figure 1) [9]. In recent years, the strongest skiers have competed without grip wax and exclusively used DP during the entire race. On race day, tracks are freshly groomed and hard packed with air temperature being −7 °C in the morning at the start (Sälen) and −3 °C when arriving at the finish line in Mora.

### 2.3. Overall Design of the Study

One week prior to the 2017 Vasaloppet race, the participant performed two roller skiing VO_2max_ ramp tests in the lab on the treadmill using diagonal stride (DIA) and the DP technique. As arterial occlusion is not applicable for core muscles and difficult to apply in limb muscles right before the start of the race with more than 15,000 other skiers in the starting area [19], maximal deoxygenation from the laboratory test was used to normalize NIRS data for the day of the race.

### 2.4. Data Collection

The two incremental roller skiing (Marwe 800 XC, wheel N-6, Marwe Oy, Hyvinkää, Finland) ramp protocols to volitional exhaustion were performed using either DIA or only DP technique to collect information about the peak physiological output (maximal HR and muscle deoxygenation). Laboratory ambient conditions were 21 °C and 24% humidity. The athlete was familiar with the performance testing settings. The protocol details can be found in Stöggl et al. [20]. In brief, the protocol consisted of a 10-min warm-up, with 6 min low-intensity (~70% HR_max_) and 4 min moderate-intensity roller skiing (~80% HR_max_), after which the ramp protocol in the respective technique began. The DP ramp protocol was performed at a fixed grade of 2°, starting at 14 km/h with an increase in speed of 1 km/h every minute until volitional exhaustion. The DIA ramp protocol had a fixed treadmill speed of 10 km/h starting at a grade of 3°, which was increased by 1° every minute until volitional exhaustion. The two ramp protocols were separated by a 20-min break involving 5-min active cool-down and 15-min passive recovery before warm-up for the second ramp test began.

During the laboratory tests, oxygen uptake (K5, Cosmed, Rome, Italy), muscle oxygenation (Moxy Monitor; Fortiori Design LLC, Hutchinson, MN, USA), and HR (Suunto Ambit Peak 3.0, Suunto, Vantaa, Finland) were monitored continuously. For VO_2_ measures, the mixing chamber mode with sampling taken every 10 sec was applied. The participant was fitted with a proper sized mask covering the mouth and nose (7450 Series V2^TM^ Mask, Hans Rudolph Inc., Shawnee, KS, USA). Prior to the test trial the gas analyzer’s oxygen (O_2_) and carbon dioxide (CO_2_) sensors were calibrated using a two calibration procedure with ambient air conditions (20.93% O_2_ and 0.03% CO_2_) and the anticipated expiratory gas percent using calibration gas containing 15% O_2_ and 5% CO_2_ (UN 1950 Aerosols, Cortex Biophysik GmbH, Leipzig, Germany) (rest volume: nitrogen). The flow volume was calibrated using a 3-L syringe (M9424, Medikro Oy, Kuopio, Finland).

Four NIRS probes were applied to important muscle groups for XCS on the right side, i.e., triceps brachii (long head at largest girth of muscle belly), latissimus dorsi (two centimeters above its origin at thoracolumbar fascia), rectus abdominus (second muscle belly from the top), and vastus lateralis (midway between lateral epicondyle and greater trochanter of femur) [13,21,22]. Skinfold thickness at the sites of the NIRS probes were 5.6 (TRI), 8.6 (LD), 8.2 (RA), and 3.4 mm (VL), respectively, determined using a Harpenden skinfold caliper (Baty International, West Sussex, UK). Positions of the NIRS devices during laboratory tests were marked with a permanent marker to place it at the same position during the race. To secure NIRS probes from moving or decoupling from the skin during the lab test and race, devices were fixed with double sided tape to the muscles, further fixed and shielded with adhesive medical tape, and secured with straps. Each of the probes weighed 40 g with dimensions of 61 mm × 44 mm × 21 mm. Neither during the laboratory tests nor the race did the participant report discomfort or inference with normal movement patterns of XCS due to the NIRS devices. To measure muscle oxygen saturation (SmO_2_), NIRS cannot distinguish between the muscular microcirculation and cytoplasm [23]. Therefore, muscle oxygenation and deoxygenation is determined as the sum of hemoglobin and myoglobin [24]. While the actual skeletal muscle oxygen uptake can be determined with the arterial (C_a_O_2_) and venous oxygen content (C_v_O_2_) derived from pulse oximetry and venous blood samples, this method was applied to limb muscles in laboratory studies [25]. As the present study involved limbs as well as core muscles, i.e., RA and LD, only SmO_2_ could be derived from oxygenated hemo-/myoglobin divided by total hemo-/myoglobin times 100 [23,24].

The Moxy Monitor applies continuous irradiation with a wavelength of 680, 720, 760, and 800 nm [23,26] with an output of mean values across the wavelengths every 0.5 s (2 Hz) [24]. Distance between the emitter and the two detectors are 12.5 and 25 mm. To account for the indefinite path length in the modified Beer–Lambert law, the Monte Carlo model is applied [24,27]. Thereby, scattering and absorption of light is based on a mixed calculation of skin, adipose, and muscle tissue [24]. Data were collected on the internal storage of each device and subsequently retrieved via Bluetooth connection. Prior to the tests, the internal clock of each NIRS probe was updated via the software (Moxy PC Application, Fortiori Design LLC, Hutchinson, MN, USA). 

During the Vasaloppet XCS race, the same four NIRS probes, HR monitor and procedures for preparation as in the laboratory were conducted. For synchronization purposes, after a countdown markers were set manually approximately 15-min prior to and 15-min after the end of the race on each probe and the HR monitor by the testers. The accuracy of this sync was <1 s and is being considered sufficient for this type of analysis and a total race time of 5 h. 

In addition, a high-precision kinematic GNSS (AT-H-02, AOBA Technologia LLC, Sendai, Japan; size: 78 mm × 38 mm × 18 mm; weight: 69 g; sampling rate: 10 Hz) was mounted on the head of the participant to achieve accurate velocity and altitude time-courses of the race [28]. 

Based on the snow-conditions and ski glide performance of the grip-waxed vs. pure glide waxed skis, the participant decided to use the strategy of exclusive DP throughout the Vasaloppet race. Directly after the end of the race the participant reported his subjective fatigue level based on a numeric rating scale (0–10).

### 2.5. Data Processing and Statistical Analysis

Before the analysis, all data were time synchronized and up-sampled to 10 Hz (equivalent to the sampling rate of GNSS) by linear interpolation. Based on the DP lab test VO_2_ data (Zone 1: ≤80% VO_2max_, Zone 2: 80–90% VO_2max_, Zone 3: ≥ 90% VO_2max_), a three-zone exercise intensity model was applied for HR and the four NIRS time series. This model was then applied during the race to achieve the exercise intensity distribution for each sensor. To compare the NIRS time series with the HR time series during the Vasaloppet race, (a) muscle deoxygenation (%, with 100% = totally deoxygenated) was calculated as 100–SmO_2_ and (b) cross-correlation procedure was applied, wherein the highest *r*-value and time-lag were reported. All results are presented as means ± standard deviation (SD). All statistical procedures and graphs were prepared using R-Studio Version 1.4.1103 (RStudio, Boston, MA, USA).

## 3. Results

Table 1 shows section times, section speeds, ranking, HR, and SmO_2_ across the eight race sections in both absolute terms relative to maximal deoxygenation as the lowest SmO_2_ measured during the all-out laboratory DP tests. Highest deoxygenation in the lab tests for all four muscles was in the range of 78–98%. Comparing skiing techniques, muscle deoxygenation tended to be higher for DIA for the upper body (TRI: 90% vs. 80%, LD: 84% vs. 78%) and trunk muscles (RA: 98% vs. 80%), while leg muscles (VL: 81% vs. 89%) showed higher muscle deoxygenation during DP.

During the Vasaloppet, the participant used the DP technique only and achieved a consistent ranking between rank 400 and 450 up to km 20 (halfway between checkpoint Smågan and Mågnsbodarna including the first and largest uphill section with approximately 200 m of gain in altitude). At this point of the race and due to high starting pace, a positive pacing strategy was visible. The distinctly decreased intensity and onset of fatigue was detected in the development of the race ranking and pronounced a decrease in the SmO_2_, especially for the RA (see Figure 1). In general, TRI and VL were the most engaged during the Vasaloppet race (67–80% and 53–59% of SmO_2_ measured in the lab at voluntary exhaustion) followed by LD (39–53%) and RA (16–65%).

In Figure 1, all sensor signals across the 90-km Vasaloppet race are illustrated. Both TRI and RA were the most engaged in the first uphill of the race (84% and 63% deoxygenation), which coincided with highest HR (182 bpm) and slowest speed. There was a marked drop in HR of 0.098 beats per minute from start to finish of the race. In contrast, both upper body muscles TRI and LD muscle deoxygenation only demonstrated a slight muscular unloading with a 0.027% drop for every minute of the race. The highest drop in muscular deoxygenation was found for RA (−0.111% per minute). The VL muscle deoxygenation remained quite stable with a drop of 0.004% and a moderate deoxygenation of ~62%. Skiing speed was stable across the race (−0.001 km/h per minute). The rate of fatigue at the end of the race was scored with a 10 (maximally fatigued).

In general, there was a comparable pattern between the HR curve and the upper-body and trunk muscle oxygenation data (see the overview in Figure 1 and zoomed illustration of undulating terrain in Figure 2). Cross-correlation analysis towards HR revealed the highest correlation for RA (*r* = 0.82), followed by TRI (*r* = 0.63) and LD (*r* = 0.56), as well as the least fit with VL (*r* = 0.35). The time-lag between HR and SmO_2_ revealed intensity changes earlier in the NIRS signal for RA (Lag = −30 s) and TRI (Lag = −2 s) followed by a later response than HR by VL (Lag = 2 s) and LD (Lag = 10 s) muscle oxygenation. The LD and RA SmO_2_ followed the fatigue pattern visible in the HR data after the first 20 km of the race (see Figure 1). However, TRI SmO_2_ reflected good changes in the terrain and speed during the entire race (for specific examples illustrating uphill and downhill sections, refer to Figure 2) but was less affected by the continuous drop visible in the HR data. Additionally, the NIRS data showed different loading patterns between upper and lower body muscles when compared with HR. In modest downhill sections, DP was still applied with high cycle velocities, which resulted in an unloading of both HR (e.g., −15 bpm) and TRI (e.g., −20%) muscle deoxygenation. However, LD activity increased or remained constant (examples from last downhill in Figure 2). During steep downhills involving long gliding sections, HR and TRI SmO_2_ increased (e.g., +20%) while leg muscle SmO_2_, i.e., VL, remained constant or even decreased (−5%). The last 15-min of the race (clear state of fatigue) demonstrated that the participant was able to increase his effort, which was observed in a slight increase of HR of about 10 bpm, while the NIRS SmO_2_ signals remained stable or even increased in magnitude (drop in muscle deoxygenation) (see Figure 3). 

Exercise intensity distribution across the 90-km Vasaloppet race based on HR and NIRS data is presented in Figure 4 and demonstrated great diversity. While HR showed a pyramidal zone distribution 81/18/1%, LD and RA exercise intensities were almost exclusively in Zone 1 (LD: 100/0/0%, RA: 90/10/0%), while TRI showed an inverse pyramidal pattern 5/24/72% and VL was mainly in Zone 2 (37/63/0%).

## 4. Discussion

The aims of the present study were to (a) investigate contribution of various upper body, lower body, and trunk muscles during a 90-km long-distance XC skiing race, (b) compare exercise intensity between HR, GNSS data, and local muscle oxygenation (NIRS signal), as well as (c) investigate robustness and non-reactiveness of wearable NIRS technology during a real race scenario. The main findings were that (1) HR basically reflected changes in terrain and intensity but showed a constant drop throughout the race; (2) TRI reflects an interchangeable pattern with HR but was less affected by drift across competition duration and may provide an alternative to HR on a local muscle level; (3) TRI and VL oxygenation revealed specific loading and unloading pattern of XCS DP in undulating terrain; and (4) RA muscle oxygenation can be particular interesting to investigate fatigue mechanisms during the locomotion of XCS.

### 4.1. NIRS to Monitor Exercise Intensity | Robustness in Field Studies

In the present study, TRI muscle oxygenation particularly changed with exercise intensity and reflected the metabolic demand of different race sections, i.e., uphill, downhill, undulating, and flat sections. RA oxygenation revealed an even quicker response and showed changes in exercise intensity 30 s before HR. Previous studies showed similar results when running in undulating terrain. Here, leg muscle oxygen saturation responded even quicker to changes in intensity than HR and reflected changes in oxygen uptake more accurately than HR [7]. As oxygen demand in the working muscle is the driving force for oxygen delivery by the cardiovascular system [29], muscle deoxygenation responded even quicker than oxygen uptake to the onset of a time trial [22]. The drift in HR across the present 90-km XCS race was also evident in previous studies that showed a drift of 0.04 bpm/km in elite and even 0.13 bpm/km in amateur skiers due to accumulating fatigue over the time course of XC skiing races [9]. Additionally, cold temperatures reduce subcutaneous blood flow, increase blood volume in internal organs, and increase stroke volume with a resulting reduction in heart rate at given exercise intensity [8]. On the other hand, HR increases 7 bpm with each 1% body weight lost due to dehydration [10]. These contrasting HR responses all may occur during a XCS race and thus impair validity of HR measurements to monitor training.

Despite changes in exercise intensity (e.g., 0.098 bpm drop per minute for HR, representing ~30 bpm across the race), TRI (0.027% per minute, respectively 8% across the race) and VL (0.004% per minute, respectively 1.2% across the race) muscle oxygenation were relatively stable throughout the race and seemed to be independent from the central fatigue mechanism that decreased HR and RA deoxygenation (0.111% per minute, almost 34% across the race). Hence, for XC skiers who are exposed to rough outdoor conditions with cold temperatures and undulating terrain, NIRS (in particular TRI and VL muscle oxygenation) may provide a feasible, non-invasive, robust, and muscle-specific method to monitor rapid changes in exercise intensity and may provide more detailed information of training potential for certain upper and lower body muscles.

### 4.2. Comparing Different Muscle Groups

Electromyography has previously been used in laboratory testing and scientific studies to develop a sophisticated understanding of complex locomotion (such as long-distance XCS), to improve movement economy, and skiing technique [30]. However, the electromyography is not yet as unobtrusive as NIRS probes, which have been used in various laboratory [13,14] and field studies to investigate exercise intensity and contribution of different muscle groups [17,18,21]. For instance, the investigation of the XCS skating technique with two NIRS applied to TRI and VL muscles showed that contribution of upper body muscles increased during flat V1/double poling and steep uphill sections. During downhill sections, VL oxygenation remained fairly stable while this rest period for upper body muscles increased TRI oxygenation [21]. Applying NIRS devices to four important muscle groups for XCS, the present study revealed large differences in local muscle intensity. For instance, RA and LD intensity was mainly in Zone 1 reflecting the central intensity distributions of HR. In contrast, TRI showed an inverse pyramidal pattern with the majority of race time working in Zone 3. While the VL was mainly in Zone 2, the VL muscle showed little alterations throughout the race despite a small decrease in SmO_2_ during steep downhill sections. Here, XC skiers engage in a crouched position with no upper body work involved but lower oxygenation of leg, i.e., VL, muscles most likely based on prolonged isometric muscle contraction with this body position. However, the incline in oxygenation of the RA muscle coincided with the onset of fatigue that was as reported by the participant and reflected in the change of the race ranking as well as reduced skiing velocity at kilometer 20. Further, the RA time course was closest related to the HR time course as reflected by the cross-correlation procedure. As RA is a major contributor for propulsion during DP [16], muscle oxygenation may serve as a fatigue marker during competition. As TRI muscle oxygenation validly displayed alternations in muscle oxygenation all throughout the race, TRI muscle oxygenation may be used to monitor training and competition load in DP during XCS.

Of special notice is the discrepancy in the exercise intensity distribution from a global (HR) versus local muscular perspective (NIRS). Distinctly different intensity distributions were observed in the selected key-indicator muscles for XCS, with heavily loaded muscle effort for TRI in an inverse pyramidal exercise intensity shape, compared with rather low intensity loading of both LD and RA and more Zone 2 loading of the VL muscles. This result might be of special interest for training control and training load on a muscular level possibly also allowing to quantify local deficits and potentials that are not visible form more macroscopic data as HR.

### 4.3. Methodological Aspects and Future Research

In order to analyze muscle oxygenation in the four most relevant muscle groups for XCS during a real race scenario, the present study was limited to one participant. As studies in elite athletes naturally come with a low number of participants with particular interest in the individual response to training and competition [31,32,33,34], equipment availability further limits the sample size in the development of new technologies [35]. For instance, Swarén and Eriksson [35] developed a power balance model to calculate propulsive power based on local positioning data of two participants during a XCS Scandinavian cup sprint race. To analyze the response of heart, muscle, kidney, and liver to ultra-endurance events in extreme temperatures (−5 to −24 °C), Niemela and Juvonen [32] analyzed a sophisticated blood marker set during the 24 h XCS world record attempt of a single skier. By applying the available NIRS devices across four muscle groups in a single skier rather than four skiers on a single muscle group, oxygenation could be compared between different muscles during the world’s most competitive long-distance XCS race. Further developments in the sensor technology might allow future possibilities to perform equal analysis during real races on a larger cohort, different skiing techniques, i.e., diagonal stride and skating, and other whole-body endurance sports, i.e., swimming, rowing, and kayaking. Although costs of NIRS devices have dramatically decreased during recent years, costs are still larger compared to HR monitors. Additionally, operator skills, correct placement, and fixation to muscle groups provide challenges to its application in the training and competition routine in a broad range of athletes.

## 5. Conclusions

The present study provides preliminary data on local muscle oxygenation measured with NIRS that reflected exercise intensity relative to two all-out laboratory efforts and quickly responded to changes in exercise intensity, i.e., changes in altitude and skiing velocity. As local exercise intensity differed between muscle groups and central exercise intensity, i.e., HR, during whole-body endurance exercise such as XCS, NIRS data measured at various major muscle groups, i.e., TRI, VL, RA, and LD, may be used for a more detailed analysis of kinetics of muscle activation during the race and compare involvement of upper body and leg muscles. The RA oxygenation may be of particular interest to investigate fatigue mechanisms during XCS races on a more microscopic level than possible with HR or GNSS data. Application of multiple NIRS sensors to different muscle groups may help to investigate weaknesses, strengths, and potentials of single skiers and develop a more individually tailored training process. During an important race, wearable NIRS applied to TRI revealed a robust and non-reactive method, and may provide a future method to monitor exercise intensity and competition load independent from fatigue mechanisms that may alter HR data. 

## Figures and Tables

**Figure 1 sensors-21-02535-f001:**
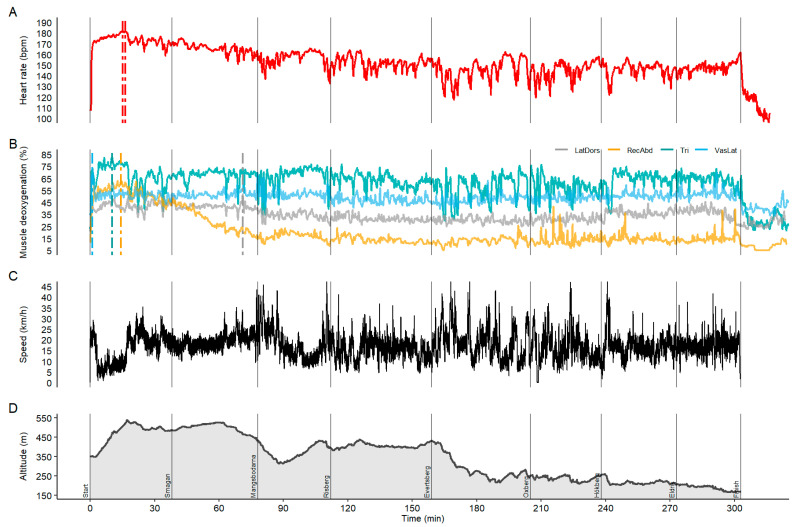
(**A**) Heart rate, (**B**) muscle oxygenation, (**C**) speed, and (**D**) altitude over the time-course and eight sections of the 90-km Vasaloppet race.

**Figure 2 sensors-21-02535-f002:**
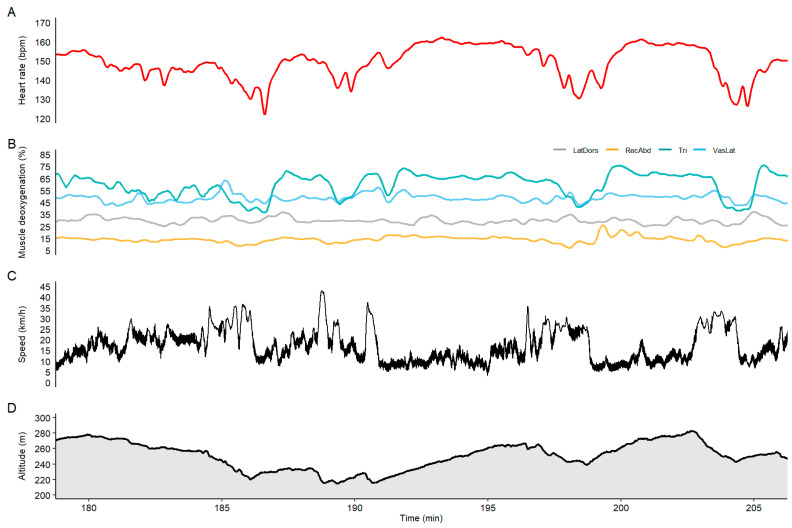
Zoomed in section of undulating terrain between Evertsberg and Oxberg, for (**A**) heart rate, (**B**) muscle oxygenation, (**C**) altitude and (**D**) speed.

**Figure 3 sensors-21-02535-f003:**
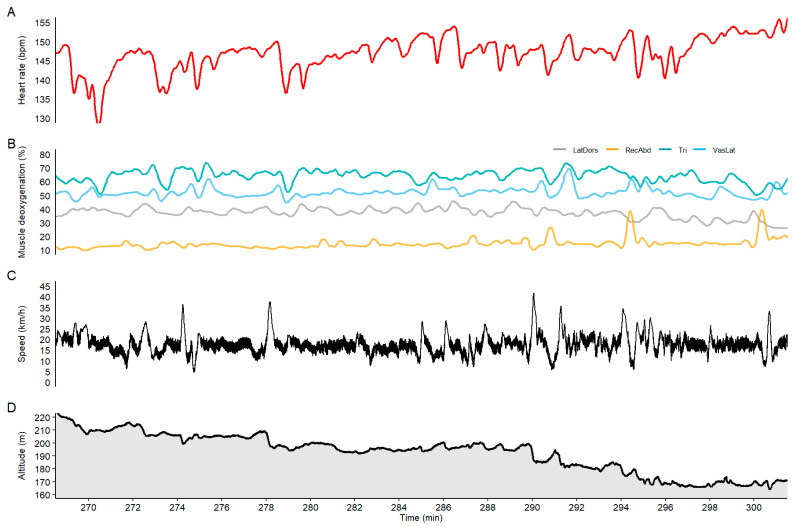
Zoomed in section of the last 10-min of the Vasaloppet race prior to the finish line for (**A**) heart rate, (**B**) muscle oxygenation, (**C**) altitude and (**D**) speed.

**Figure 4 sensors-21-02535-f004:**
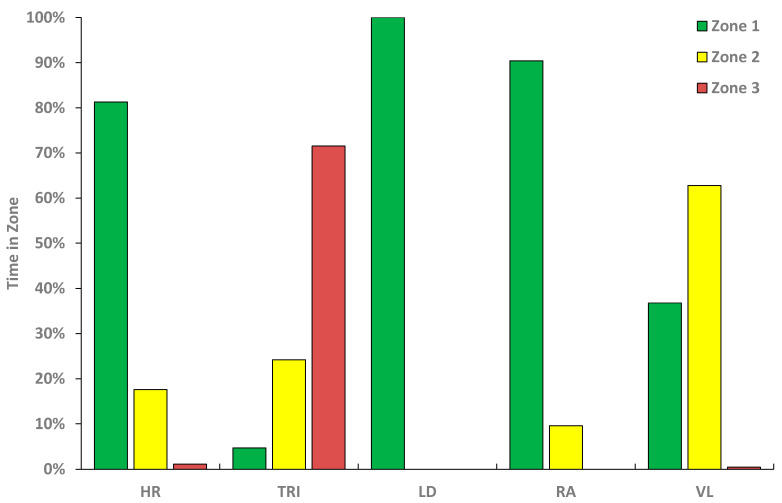
Exercise intensity distribution during the 90-km Vasaloppet based on heart rate (HR) data and muscle oxygenation of triceps brachii (TRI), latissimus dorsi (LD), rectus abdominis (RA), and vastus lateralis (VL). The three zones are according to the respective values during the laboratory DP ramp protocol based on the VO_2_ data (Zone 1: ≤80% VO_2max_, Zone 2: 80–90% VO_2max_, Zone 3: ≥ 90% VO_2max_).

**Table 1 sensors-21-02535-t001:** Sectional analysis of split times, ranking, mean ± SD for section speed, heart rate (HR), and muscle deoxygenation of triceps brachii (TRI), latissimus dorsi (LD), rectus abdominis (RA), and vastus lateralis (VL) during the 90-km Vasaloppet in absolute values and relative to the peak deoxygenation and peak HR during the laboratory roller skiing ramp tests involving diagonal stride (DIA) and double poling (DP).

		Time (h:mm:ss)	Ranking	Speed (km/h)	HR (bpm)	TRI (%)	LD (%)	RA (%)	VL (%)
	Laboratory Tests								
Peak DP–Laboratory					188	85	78	80	89
Peak DIA–Laboratory					187	90	84	98	81
	**90-km Vasaloppet**								
Start: Smågan		0:38:06	410	16.3 ± 7.1	173 ± 9 92.4%	68 ± 10 79%	42 ± 4 53%	52 ± 7 65%	52 ± 4 58%
Smågan-Mångsbodarna		1:18:00	439	19.9 ± 3.9	166 ± 4 88.7%	68 ± 5 80%	42 ± 3 53%	30 ± 9 37%	52 ± 3 58%
Mångsbodarna-Risberg		1:52:25	482	18.6 ± 7.6	157 ± 6 83.9%	67 ± 9 78%	34 ± 3 44%	17 ± 3 21%	52 ± 4 58%
Risberg-Evertsberg		2:39:06	597	16.1 ± 5.1	152 ± 6 81.4%	64 ± 5 74%	32 ± 2 40%	14 ± 3 17%	47 ± 3 53%
Evertsberg-Oxberg		3:24:56	649	19.2 ± 8.0	146 ± 10 77.9%	57 ± 11 67%	30 ± 2 39%	13 ± 3 16%	47 ± 4 53%
Oxberg-Hökberg		3:58:21	720	16.3 ± 7.1	146 ± 8 77.8%	58 ± 8 68%	31 ± 3 40%	14 ± 5 17%	49 ± 3 55%
Hökberg-Eldris		4:32:56	746	17.6 ± 5.6	146 ± 6 78.1%	64 ± 8 75%	36 ± 3 46%	14 ± 3 18%	51 ± 3 57%
Eldris–Finish		5:02:40	750	17.8 ± 4.6	148 ± 4 79.1%	64 ± 5 75%	37 ± 4 47%	15 ± 4 19%	52 ± 4 59%
Change over time from Start to Finish (units/min)				−0.001	−0.098	−0.027	−0.027	−0.111	−0.004

## Data Availability

Not Applicable.

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
