# Peer review of "Near Infrared Spectroscopy for Muscle Specific Analysis of Intensity and Fatigue during Cross-Country Skiing Competition—A Case Report"

_sensors, 2021, doi:10.3390/s21072535_

Round 1

Reviewer 1 Report

OVERALL

The draft presents a dataset from a case study on a potentially novel application; cross-country skiing (XCS), of NIRS based measurements of muscle oxygenation as a proxy of exercise intensity/endurance. There are precedents in the literature regarding the use of NIRS based measurements of muscle oxygenation regarding a related skiing application (roller skiing) but as far as I know, this is the first in XCS. There, I think lies the major value of this report.

However, the collection settings, data processing and analysis and reporting are somewhat below what one would expect from a polished scientific report, where much more precision in the description of methods is expected so to guarantee replicability, and objective quantitative analysis should, in general, be favoured over subjective appreciation on trends (more on this below).

The suitability for Sensors can also be questioned. I am not certain about Sensors tolerance for case studies, and even if these are accepted, this draft is not so much about sensing but more about the application. There are journals dedicated to sports science that may be perhaps more suitable.

Yet despite all my heavy criticism here, all in all, I enjoyed the draft and so I would be happy to see my current rejection recommendation turned during a potential rebuttal.

SUGGESTIONS TO IMPROVE THE PAPER

Major:

+ The authors appear to mix concepts such as exercise intensity, fatigue and endurance. Although closely related, they are and should be treated as different constructs. This has important implications not only for the conclusions and interpretation but also for how the experiment is conducted. For instance, should exercise endurance be the target, then concomitant measurements of the rate of oxygen uptake (VO2) -at least during the pre-test if not during the race itself should have been collected. Analogously, if fatigue was targeted than the Sam-Pirelli scale or de Fatigue severity scale is often reported, and if it was exercise intensity VO2 during a graded exercise testing or measuring the anaerobic threshold via the blood lactate concentration would be suitable gold standards. My point is simple, without clarity on the construct nor companion gold standard measurements, It is difficult to establish whether NIRS based measurements of muscle oxygenation are in this case acting as a good proxy of exercise intensity/endurance. At most, only a correlation among timecourses can be established.

+ This takes me to my second observation, there isn’t any correlation among time courses, neither global nor at local intervals of interest, nor the timecourses during the preliminary tests are reported (except for the entry in Table 1).

+ In ln 53, it seems that 3 chromophores are to be decoded; HbO2, HbR and myoglobin, but the methods (ln 133) suggests that only 2 wavelengths i.e. two parameters, will be decoded. I suppose HbO2 and HbR, but this is not clear. Also, how was Hb data reconstructed? Using the (modified) Beer-Lambert law? Any DPF correction? What is the underpinning tissue model? Which emitter-detector separation? Which power, gain and calibration?

+ Marking sensor target location with a marker does not prevent the optical sensor from moving and decoupling from the skin, especially during sports activities. How was consistent placement enforced?

+ Report of statistics is deficient; Analysis is restricted to descriptive statistics, and even that is very limited. it is indicated that std is given, but I can’t find it. All statements of relations among variables are subjective and are not established using any mathematical models.

+ More on imprecision. I assume that when the authors speak of muscle oxygenation they refer to SO2= 100*cHbO2/cHbT, but this is not indicated. Since neither oxygen plasma, blood flow nor physiological gases have been measured, it is not possible to determine oxygen content CaO2 or oxygen consumption CMRO2.

+ As appreciated from the figures, time traces of the muscle oxygenation are clearly affected by motion artefacts. How was data processed to correct for motion artefacts?

+ The authors put some emphasis on the relation between HR and muscle oxygenation. While Fick’s equation relates both concepts, this relation is non-linear [Beltz et al 2016], basically because skeletal muscles have a large capacity for increasing blood flow which greatly exceeds the heart pumping capacity [Sultin et al 1985]. Moreover, this dissociation is more pronounced in athletes. With these known precedents, why would the authors expect a strong relationship between these two physiological signals? What’s the rationale?

Minor:

+ I assume the sensor works in continuous-wave irradiation but again, this information is not given.

+ It is likely that the subject is Caucasian but this is not indicated. Since skin tone affects oxygenation measurements, even if calibration for dark skins is trivial, this information is needed for correct interpretation.

+ Which algorithm was used for upsampling?

+ How was skin thickness calculated? I can assume that this was done using callipers, but this is not indicated.

+ It is indicated that signals were synchronized at the beginning, but the length of the task likely requires intermediate markings. Was this attempted? If not what are the guarantees that a single initial marking is sufficient?

+ Even although originally developed for clinical purposes, but it may be convenient to follow (an adapted version) of the CARE standard for reporting case studies.

Author Response

Overall

The draft presents a dataset from a case study on a potentially novel application; cross-country skiing (XCS), of NIRS based measurements of muscle oxygenation as a proxy of exercise intensity/endurance. There are precedents in the literature regarding the use of NIRS based measurements of muscle oxygenation regarding a related skiing application (roller skiing) but as far as I know, this is the first in XCS. There, I think lies the major value of this report.

However, the collection settings, data processing and analysis and reporting are somewhat below what one would expect from a polished scientific report, where much more precision in the description of methods is expected so to guarantee replicability, and objective quantitative analysis should, in general, be favoured over subjective appreciation on trends (more on this below).

The suitability for Sensors can also be questioned. I am not certain about Sensors tolerance for case studies, and even if these are accepted, this draft is not so much about sensing but more about the application. There are journals dedicated to sports science that may be perhaps more suitable.

Yet despite all my heavy criticism here, all in all, I enjoyed the draft and so I would be happy to see my current rejection recommendation turned during a potential rebuttal.

A: We appreciate the reviewer’s time and effort to undertake the manuscript such a detailed assessment. The reviewer is clearly an expert in the field and the very constructive comments helped to hopefully improve the manuscript substantially.

Major comments

+ The authors appear to mix concepts such as exercise intensity, fatigue and endurance. Although closely related, they are and should be treated as different constructs. This has important implications not only for the conclusions and interpretation but also for how the experiment is conducted. For instance, should exercise endurance be the target, then concomitant measurements of the rate of oxygen uptake (VO2) -at least during the pre-test if not during the race itself should have been collected. Analogously, if fatigue was targeted than the Sam-Pirelli scale or de Fatigue severity scale is often reported, and if it was exercise intensity VO2 during a graded exercise testing or measuring the anaerobic threshold via the blood lactate concentration would be suitable gold standards. My point is simple, without clarity on the construct nor companion gold standard measurements, it is difficult to establish whether NIRS based measurements of muscle oxygenation are in this case acting as a good proxy of exercise intensity/endurance. At most, only a correlation among timecourses can be established.

A: Thank you for these comments. While reading the comments we clearly need to agree that concepts were mixed and not perfectly described. During baseline testing we had collected oxygen uptake in both lab-tests. As we were not allowed to collect VO2 data during the actual race we did not present these information in the first submission. However, based on your comments we fully agree that these data need to be added and used the VO2 data for an exercise intensity distribution analysis. The time points of 80% and 90% of VO2max were used to build up a three zone model separately for HR and the four NIRS data sets (e.g. HR at 80%VO2max, HR at 90% VO2max, SmO2 TRI at 80% VO2max, etc.). We have now a time in zone plot for HR and the 4 NIRS signals (See new Figure 4 and new text on P9, L255-259).

With respect to fatigue – unfortunately we have not collected subjectively reported fatigue state (E.g. fatigue scale or Sam-Pierelli) during the race. We only assessed the state of fatigue (1-10 fatigue scale) at the finish line and documented a subjective race description of the participant. E.g. good race performance until km 20, then needed to reduce speed based on overpacing.

Please find the details further below under “specifics”.

+ This takes me to my second observation, there isn’t any correlation among time courses, neither global nor at local intervals of interest, nor the timecourses during the preliminary tests are reported (except for the entry in Table 1).

A: * Thank you for this input. We have now added the information about VO2 measures and VO2max during DIA and DP in the lab tests. Further, we have calculated cross-correlations between the single signals to check for “similarity” and/or phase shifts in the signals. We have added the highest r-values from the cross-correlation and the time-lag between HR and the four NIRS signals. See Results P7, L 231-233.

+ In ln 53, it seems that 3 chromophores are to be decoded; HbO2, HbR and myoglobin, but the methods (ln 133) suggests that only 2 wavelengths i.e. two parameters, will be decoded. I suppose HbO2 and HbR, but this is not clear. Also, how was Hb data reconstructed? Using the (modified) Beer-Lambert law? Any DPF correction? What is the underpinning tissue model? Which emitter-detector separation? Which power, gain and calibration?

A: We totally agree and provided more details on the specifics of the NIRS technology used. Please refer to L137-165.

+ Marking sensor target location with a marker does not prevent the optical sensor from moving and decoupling from the skin, especially during sports activities. How was consistent placement enforced?

A: Positions of the NIRS devices during laboratory tests were marked with a permanent marker to place it at the same position during the race. To secure NIRS probes from moving or decoupling from the skin during skiing action, devices were a) the bottom area of the sensor fixed with double-sided tape (provided by MOXY) directly on the skin , b) further fixed and shielded with adhesive medical tape and c) secured with straps. Further, the elastic and tight XCS race suite guaranteed good additional fixture. We added this to the methods (L143-147f).

+ Report of statistics is deficient; Analysis is restricted to descriptive statistics, and even that is very limited. It is indicated that std is given, but I can’t find it. All statements of relations among variables are subjective and are not established using any mathematical models.

A: Please find the new statistics about cross-correlations and the exercise intensity distribution.

Further, we have added the SD values in the Table 1.

+ More on imprecision. I assume that when the authors speak of muscle oxygenation they refer to SO2= 100*cHbO2/cHbT, but this is not indicated. Since neither oxygen plasma, blood flow nor physiological gases have been measured, it is not possible to determine oxygen content CaO2 or oxygen consumption CMRO2.

A: Thank you for your input. We revised the description of the NIRS methodology. We indicated that muscle oxygen saturation (SmO2) was determined by oxygenated hemo-/myoglobin divided by total hemo-/myoglobin (as the sum of oxygenated and deoxygenated hemo-/myoglobin) times 100. While the actual skeletal muscle oxygen uptake based on the arterial (CaO2) and venous oxygen content (CvO2) derived from pulse oximetry and venous blood samples may be studied in limb muscles this method is difficult to apply in core muscles, i.e. rectus abdominus and vastus lateralis, especially when measuring in the field. We clarified this accordingly (L150f).

+ As appreciated from the figures, time traces of the muscle oxygenation are clearly affected by motion artefacts. How was data processed to correct for motion artefacts?

A: Although NIRS signals collected during whole-body activity in particular real-race scenario are not as clean as data derived from single limb activity such as hand grip exercises derived from laboratory studies, we could not identify motion artefacts. Therefore, no further correction or smoothing was applied. Further, based on the ~ 1Hz to maximal 1.5 Hz cycle duration during XCS a motion artefact correction at a sampling rate of 2 Hz is not plausible.

+ The authors put some emphasis on the relation between HR and muscle oxygenation. While Fick’s equation relates both concepts, this relation is non-linear [Beltz et al 2016], basically because skeletal muscles have a large capacity for increasing blood flow which greatly exceeds the heart pumping capacity [Sultin et al 1985]. Moreover, this dissociation is more pronounced in athletes. With these known precedents, why would the authors expect a strong relationship between these two physiological signals? What’s the rationale?

A: You are absolutely right. Based on Fick’s equation, muscle oxygen uptake can be derived from blood flow, arterial and venous oxygen content {Carr, 2019 #607}. However, with the present NIRS methodology, only muscle oxygen saturation (SmO2) can be determined from oxygenated and deoxygenated hemo-/myoglobin {Feldmann, 2019 #600}{McManus, 2018 #500}{Scholkmann, 2020 #601}. While HR is generally used in the field to monitor exercise intensity, it is not considered the gold standard. Indeed, previous studies showed that there is a dissociation between HR and VO2 in particular during race simulations with continuously changing intensity due to steep uphill and downhill sections {Born, 2017 #609}. Therefore, the use of NIRS was suggested to provide an alternative solution to monitor exercise intensity in such conditions. Additionally, in previous studies an inflection point in the tissue oxygen saturation and muscle deoxygenation was detected that were correlated to intensity at maximal lactate steady state (r = 0.92) {Snyder, 2009 #611} and onset of blood lactate concentration (r2 = 0.95), respectively {Grassi, 1999 #614}. However, during a real-race scenario such as long-distance XCS, neither lactate nor VO2 measurements are possible (or hardly to be established if an athlete really wants to “race”). Lactate samples would require rest periods during the race while most of the Vasaloppet track is difficult to access from an outside person. VO2 measurements are limited by battery life of the gas analyzer and rules of race organizers prohibiting the use of breathing masks that are connected to an electronic medical to avoid potential performance enhancing regimes. Therefore, in the present study, relationships between SmO2, VO2, and HR were investigated and exercise intensity distribution derived from lab-tests. We added this to the methods, results and discussion as mentioned above.

Minor comments

+ I assume the sensor works in continuous-wave irradiation but again, this information is not given.

A: We added the missing information to the specification of the NIRS methodology used in the present study L150ff

+ It is likely that the subject is Caucasian but this is not indicated. Since skin tone affects oxygenation measurements, even if calibration for dark skins is trivial, this information is needed for correct interpretation.

* We indicated the subjects race in the methods. L88

+ Which algorithm was used for upsampling?

A:  The data was upsampled by lineare interpolation. L184

+ How was skin thickness calculated? I can assume that this was done using calipers, but this is not indicated.

A: You are right. Skinfold thickness was measured using a Harpenden skinfold caliper. We indicated this accordingly. L 142-143

+ It is indicated that signals were synchronized at the beginning, but the length of the task likely requires intermediate markings. Was this attempted? If not what are the guarantees that a single initial marking is sufficient?

A:  This is an important point. Unfortunately, the single sensors do not communicate with each other to provide perfect synchronization. However, each sensor has a sync button. So prior to start the 4 sync buttons + the HR monitor were pressed by the testers following a count down. The same procedure was done  15 min after the race, before systems were switched off. In addition, each sensor has a timestamp. We did not explore any marked drift between all 6 sensors, and the time courses of the signals seemed to nicely resemble the track profile. In our opinion, this method allows at least a synchronization with accuracy of ~1 s. Text was added on page 4, L 170-172.

+ Even although originally developed for clinical purposes, but it may be convenient to follow (an adapted version) of the CARE standard for reporting case studies.

A: Thank you for this comment. We have checked the care standards and tried to integrate possible points (e.g. Title)

Reviewer 2 Report

The ability to effectively monitor athlete exertion is an area of interest in exercise physiology and sports medicine. The increasing availability of technological devices at decreasing costs requires the experimentation of new solutions. In this perspective, near-infrared spectroscopy (NIRS) in particular for its non-invasiveness, flexibility of use and selectivity of study in specific dynamic conditions deserves particular attention.

The Authors in a single-case study report on the use of a wearable device based on near-infrared spectroscopy (NIRS) technology to monitor exercise intensity during a cross country race in a former athlete. The data are compared with heart rate (HR) monitoring and high-precision global navigation satellite (GNSS)  data

The Authors’ conclusions confirm the suitability of NIRS  to monitor exercise intensity and muscle activity in an outdoor setting  and to represent a possible alternative method to HR and GNSS for exercise monitoring.   

Main comment

The topic is interesting and the technology applicable. The study is well conducted and the manuscript is well and accurately written. However, the critical factor in the manuscript is the limited sample size. It is a single case collected from an ultra-endurance race in a sport involving the arms. These aspects together make the study preliminary, which should be noted throughout the manuscript.

Comments

  • Title: this is a case report and this need to be stated in the title (e.g. …cross-country skiing competition: a case report or a single-case study

  • Abstract section acronyms HR, VL and RA needs to be expanded to enhance clarity.
  • Methods
    - The number of subjects should be Lines 82 to 86 are not reflecting the methods of the study; I suggest to move them to discussion   section (limitations).

  • Please insert the distance between NIR-light sources and detectors, to identify a theoretical depth of light penetration
  • Some information about the applicability and the feasibility of the NIRS measurements during a competitive race may be added. Did the sensors interfere with the athlete performance?

  • I have some concerns about the calculation of the deoxygenation during the lab test (line 157). Authors express the deoxygenation as a percentage, but they need to clarify how did they calculate it. Is it a oxygen saturation? Or is it a percentage of the resting value? This need to be explained to help the data interpretation.  

  • Authors need also to clarify if they calculated the percentage of deoxygenation on the traces of oxyhemoglobin or deoxyhemoglobin.

  • Table 1: the average speed for each fraction should be also added

  • The relationship between HR% and TRI% has not been considered. I found  R2 =0.64  with  consistency between parameters  as assessed by the Bland-Altman plot. This fact should be considered   when discussing the necessity of an alternative use of NIRS instead of HR.  

  • Discussion
  • NIRS trace vs HR: The comment on the relationship between parameters should be considered.

The Authors should clarify the specific benefits of using NIRS considering that methods  have differences and limitations, as well different costs and operator’s skills required.   Specific disciplines on which to focus new studies  might be suggested on the basis of this experience.

-              A limitation section is missing and since this is a case-report, this at least need to stated, with  possible related comments   (P2 L82-86 )

Author Response

Overall

The ability to effectively monitor athlete exertion is an area of interest in exercise physiology and sports medicine. The increasing availability of technological devices at decreasing costs requires the experimentation of new solutions. In this perspective, near-infrared spectroscopy (NIRS) in particular for its non-invasiveness, flexibility of use and selectivity of study in specific dynamic conditions deserves particular attention.

The Authors in a single-case study report on the use of a wearable device based on near-infrared spectroscopy (NIRS) technology to monitor exercise intensity during a cross country race in a former athlete. The data are compared with heart rate (HR) monitoring and high-precision global navigation satellite (GNSS) data.

The Authors’ conclusions confirm the suitability of NIRS to monitor exercise intensity and muscle activity in an outdoor setting  and to represent a possible alternative method to HR and GNSS for exercise monitoring.

A: We appreciate the reviewer’s assessment. We feel that the valuable comments made helped us to improve the manuscript.

Main comments

The topic is interesting and the technology applicable. The study is well conducted and the manuscript is well and accurately written. However, the critical factor in the manuscript is the limited sample size. It is a single case collected from an ultra-endurance race in a sport involving the arms. These aspects together make the study preliminary, which should be noted throughout the manuscript.

A: We clarified that the present manuscript describes a preliminary case report.

Title

This is a case report and this need to be stated in the title (e.g. …cross-country skiing competition: a case report or a single-case study

A: Added accordingly.

Abstract section acronyms HR, VL and RA needs to be expanded to enhance clarity.

A: Revised accordingly.

Methods

The number of subjects should be Lines 82 to 86 are not reflecting the methods of the study; I suggest to move them to discussion   section (limitations).

A: We now discuss number of subjects in the ‘methodological considerations’ (L344ff).

Please insert the distance between NIR-light sources and detectors, to identify a theoretical depth of light penetration

A: We specified that the distance between emitter and the two detectors was 12.5 and 25 mm (L162).

Some information about the applicability and the feasibility of the NIRS measurements during a competitive race may be added. Did the sensors interfere with the athlete performance?

* Neither during the laboratory tests nor the race, the participant reported discomfort or inference into the normal movement pattern of XCS due to the NIRS devices. We added this accordingly (L148f).

I have some concerns about the calculation of the deoxygenation during the lab test (line 157). Authors express the deoxygenation as a percentage, but they need to clarify how did they calculate it. Is it a oxygen saturation? Or is it a percentage of the resting value? This need to be explained to help the data interpretation.

A: we have added more specific information about the NIRS probes and the calculated metrics for SmO2. In order to better compare with the shape of the HR curve we have taken 100- SmO2 for muscle deoxygenation. Therefore, an increase in the load is resulting in an increase in HR and also an increase in the deoxygenation. See P4, L144ff and L184

Authors need also to clarify if they calculated the percentage of deoxygenation on the traces of oxyhemoglobin or deoxyhemoglobin.

* You are right. We clarified that muscle oxygen saturation (SmO2) was calculated from oxygenated and deoxygenated hemo-/myoglobin as the NIRS technology cannot distinguish between oxygen content in muscular microcirculation (hemoglobin) and cytoplasm (myoglobin). We revised the manuscript accordingly.

Table 1: the average speed for each fraction should be also added

*A: Thank you for your comment. We have added this information in Table 1.

The relationship between HR% and TRI% has not been considered. I found R2 =0.64 with consistency between parameters as assessed by the Bland-Altman plot. This fact should be considered when discussing the necessity of an alternative use of NIRS instead of HR.

* A: Thank you for this information. We have now added cross-correlations between HR and NIRS signals. Highest correlation was found for RA followed by TRI and LD and least with VL. See new text in results. L 231-235.

Discussion

NIRS trace vs HR: The comment on the relationship between parameters should be considered.

A: Thank you. We have added text in the discussion with respect to cross-correlations but also the exercise intensity distribution according to HR vs. NIRS data. See L282-283, 318-323, 328-329, 334ff

The Authors should clarify the specific benefits of using NIRS considering that methods have differences and limitations, as well different costs and operator’s skills required. Specific disciplines on which to focus new studies might be suggested on the basis of this experience.

A: We discussed the specific benefits and limitations of NIRS compared to HR accordingly – see also comments above (L307f).

A limitation section is missing and since this is a case-report, this at least need to stated, with possible related comments (P2 L82-86)

A: We discussed limitations and the case report issue in ‘methodological considerations’. L344 ff.

Round 2

Reviewer 1 Report

As I indicated in my original review, I was hoping for a good rebuttal to turn my initial rejection recommendation, and I got it. The authors have thoroughly revised the paper to address all of my initial concerns. I am very happy to recommend this work for publication. 

Reviewer 2 Report

The Authors have addressed all the comments of the reviewer.